# DiDI: Disentangle Denoise Inject for Improving T2I Diffusion Models

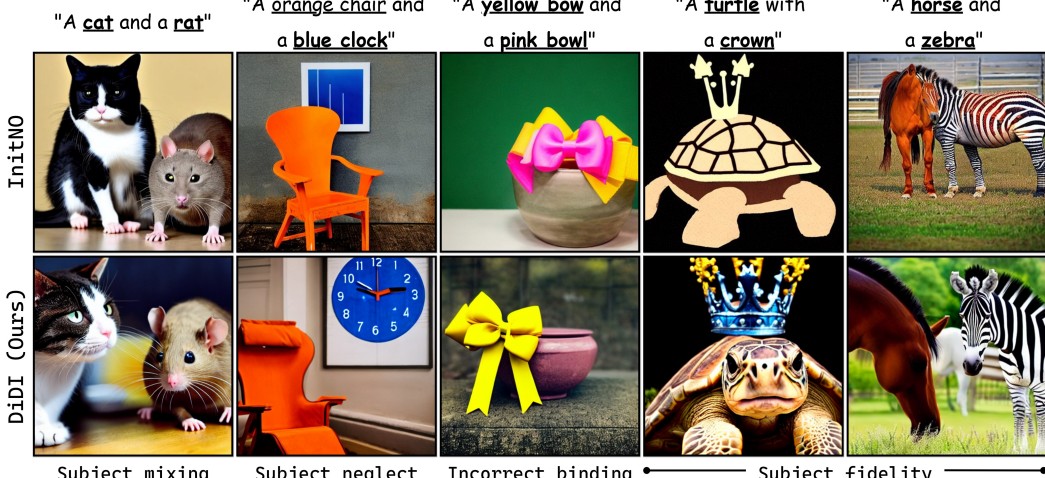

Figure 1: Images generated by InitNO (Guo et al., 2024) and ours. The proposed approach mitigates common T2I alignment failures in pre-trained SD and existing SOTA methods.

## ABSTRACT

Text-to-image (T2I) synthesis has been revolutionized by diffusion models. However, state-of-the-art (SOTA) like Stable Diffusion still suffers from well-known alignment issues like subject mixing and subject neglect when composing from multi-subject prompts. While recent efforts aim to address these misalignment issues, they remain vulnerable to bad initial noise seeds. We propose Disentangle Denoising Inject (DiDI): a training-free pipeline that improves T2I alignment even under poor initializations, by explicitly modifying the initial latent representation. It is observed that the alignment failures in multi-subject settings, caused by cross-attention overlap and inadequate subject semantics fundamentally stem from a poor initial noise. To this end, DiDI performs spatial disentanglement on the initial latent to mitigate subject mixing. Our core insight is that diffusion models, despite sub-optimal seeds, can still reliably synthesize *individual* subjects and their attributes. Accordingly, DiDI introduces a partial denoising scheme to generate early semantic features for individual subjects and attributes. By injecting representative features into the disentangled latent, DiDI successfully guides the denoising process towards more faithful generations. Extensive experiments demonstrate the superiority of our method compared to existing SOTA approaches.

## 1 INTRODUCTION

Text-to-image (T2I) synthesis has achieved remarkable progress, driven by advances in diffusion models like Stable Diffusion (SD) (Rombach et al., 2022; Podell et al., 2023). These models far outperform their predecessors like GANs (Tao et al., 2022; Liao et al., 2022) in producing highly

realistic and artistic images across different domains. Despite their strengths, diffusion models struggle to adhere to text prompts due to misalignment issues. Common failures including subject mixing (Qiu et al., 2025), subject neglect (Chefer et al., 2023; Marioriyad et al., 2024), and incorrect attribute binding, still persist in SOTA models (Guo et al., 2024) as shown in Fig. 1.

Recent studies trace the causes of misalignment to the initial noise latent, underscoring its influence on the final output. Consequently, some methods mine for good seeds (Mao et al., 2024; Xu et al., 2025), or optimize the initial noise (Guo et al., 2024) for faithful generations. However, such favorable seeds are rare leaving most generations to sub-optimal alignment. Besides, extensive seed sampling is computationally expensive and generalizes poorly to different compositions like animals, objects, and colors. Similarly, optimizing a denoising process with a bad seed is inherently challenging due to several underlying causes such as poor spatial layouts (Li et al., 2024a; Marioriyad et al., 2024) and inadequate attention scores. Above all, these methods are limited by the biases of the text encoder e.g., CLIP (Radford et al., 2021). As shown in previous studies (Abbasi et al., 2025; Xie et al., 2025), CLIP tends to prioritize early subjects and produce entangled representations when captions involve multiple subjects. This hinders the learning of individual concepts.

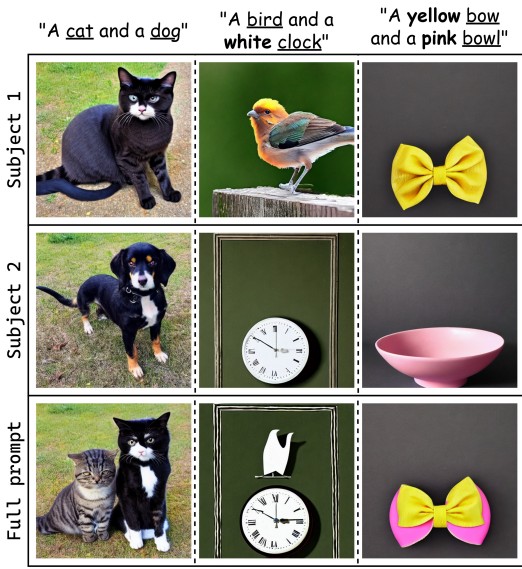

Figure 2: SD fails to compose multiple subjects but successfully renders them when the prompt is decomposed into individual subjects and **attributes**, using the same seed.

Generally, early denoising steps remove high-frequency noise and organize the semantics of high-level concepts (Brack et al., 2023); thus, multi-subject synthesis can be viewed as constructing a coherent scene from independently modeled subjects. We hypothesize that improving subject generation at the earliest denoising stage will strengthen T2I alignment. To this end, we propose a novel framework Disentangle Denoising Inject (DiDI): a training-free pipeline that modifies the initial noise sample to encourage faithful generation under multi-subject settings.

DiDI consists of three core processes: (1) initial latent spatial disentanglement, (2) subject-aware partial denoising, and (3) semantic feature injection, designed to address subject mixing and subject neglect. Our analyses of attention maps reveal that subject mixing mainly occurs due to cross-attention overlap, which appears as early as the first denoising timestep. To mitigate this, DiDI disentangles the initial latent by maximizing the distance between the subjects' centers of mass, thereby optimizing their spatial arrangement.

However, disentanglement alone fails to resolve two key issues: (1) inadequate generation, where the latent lacks sufficient semantic information for each subject, and (2) misrepresented semantics, where similar subjects mix due to entangled text embeddings. Naïve latent editing introduces artifacts, while fully generating and merging individual subjects is computationally expensive. To address this, DiDI leverages a subject-aware partial denoising strategy. This is based on our observation that diffusion models, while challenged by multi-subject generation, can reliably synthesize a single subject and its attributes (see Fig. 2). Each subject is partially denoised to extract characteristic semantic features, which are then injected into the disentangled latent through semantic feature injection. This mitigates subject neglect and enhances subject fidelity. Interestingly, the results also show that good injections lead to more accurate attribute binding.

Our work provides a new perspective on the role of the initial noise in T2I misalignments, and introduce a framework for editing this noise towards coherent multi-subject generation. DiDI is a plug-and-play pipeline, which consistently outperforms vanilla SD, initial noise-based methods (Guo et al., 2024), and SOTA test-time optimization methods (Qiu et al., 2025) in generating semantically accurate images.

## 2    RELATED WORK

**T2I Diffusion Models.** Diffusion models (Ho et al., 2020; Nichol & Dhariwal, 2021; Dhariwal & Nichol, 2021) addressed the limitations of previous models in reproducing the variability of real images by modeling image distributions through a parameterized Markov chain. However, early diffusion models like denoising diffusion probabilistic models (DDPMs) (Ho et al., 2020) struggled with poor T2I alignment and high computational requirements. The departure to latent space (Rombach et al., 2022) along with efficient samplers and schedulers (Song et al., 2020; Xu et al., 2024; Duan et al., 2023) mitigated the latter shortcomings. Classifier guidance (Dhariwal & Nichol, 2021) helped with T2I alignment by guiding the denoising process according to the class gradients of a classifier. Later, classifier-free guidance (Ho & Salimans, 2022) achieved similar performance using both conditional and unconditional diffusion models. Current models like SD v3 (Esser et al., 2024) and DALL-E 3 (Betker et al., 2023) improve alignment by using multiple encoders for conditioning, but these models require retraining that is costly and limits their application. Despite attempts like prompt engineering (Fan et al., 2024; Wang et al., 2024; Li et al., 2024b), well-studied issues such as subject neglect, subject mixing, and incorrect attribute binding remain open challenges.

**Test-time Optimization.** An active field of study involves training-free methods, which manipulate the denoising process during inference to improve T2I alignment. Some approaches (Dahary et al., 2024; Feng et al., 2023; Lian et al., 2023) use large language models (LLMs) and vision language models (VLMs) for layout supervision. While these approaches are effective in controlling a generation process, they rely on external knowledge and often fail to resolve core issues like subject neglect and attribute binding. Other works avoid external supervision, e.g., Composable Diffusion (Liu et al., 2022) composes models via a product-of-experts, but scales poorly with prompt complexity. Methods like Attend-and-Excite (Chefer et al., 2023), MaskDiffusion (Zhou et al., 2025), CONFORM (Meral et al., 2024), and Self-Cross (Qiu et al., 2025) leverage the attention maps to alleviate subject mixing and neglect. Yet, all these methods are confined by the number of inference steps. Moreover, they are vulnerable to bad initial seeds having poor generative properties. In contrast, DiDI directly edits the initial noise, and can be paired with existing test-time approaches to further enhance performance.

**Initial Noise Optimization.** Prevailing studies (Mao et al., 2023; Samuel et al., 2024) show that regions in the initial noise influence the generation of specific content, and editing this noise significantly impacts the final image. For instance, Everaert et al. (2024) balanced the low-frequency components in the noise sample to correct artifacts related to brightness. InitNO (Guo et al., 2024) optimized the noise for better semantic alignment. Other approaches (Mao et al., 2023) manipulate the latent for layout-guided synthesis and image repainting tasks. FIND (Chen et al., 2024) introduced a policy optimization framework to shape its noise distribution towards a desired content. Additionally, methods involving reliable seed mining (Li et al., 2024a) have revealed strong correlations between the initial noise and generation fidelity, highlighting its critical role in diffusion models.

## 3    PRELIMINARIES

**Text Guided Image Synthesis.** Text guidance in SD takes place in the cross-attention layers of the denoising U-Net (Ronneberger et al., 2015). The CLIP embeddings $\mathbf{c} = \{c_1, \ldots, c_N\}$ of a text prompt with $N$ tokens $\mathbf{y} = \{y_1, \ldots, y_N\}$ are projected to the keys $K$ and values $V$, while the intermediate latents $\mathbf{z}_t$ from the U-Net are projected to queries $Q$ as:

$$Q = W_Q \cdot z_t, \ K = W_K \cdot c, \ V = W_V \cdot c. \tag{1}$$

Given a pre-trained network with parameters $W$, the cross-attention map is computed as the scaled dot product between $Q$ and $K$:

$$\mathrm{Attention}(Q, K, V) = \mathrm{softmax}\left(\frac{QK^T}{\sqrt{d}}\right)V, \tag{2}$$

where $d$ is the number of channels. The cross-attention map for a token $y_i$ is denoted by $A_{\mathrm{ca}}^{y_i} \in \mathbb{R}^{w/P \times h/P}$, where $w$ and $h$ are the latent dimensions, and $P$ is the patch size. This map defines a distribution over $(w/P \times h/P)$ spatial patches, highlighting the regions of the latent that are

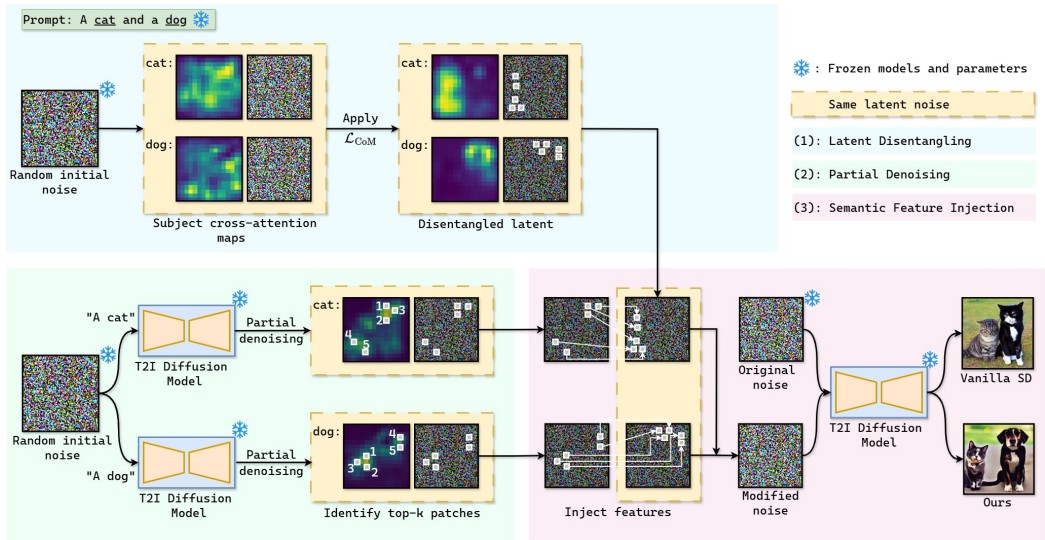

Figure 3: Overview of the DiDI pipeline. DiDI mitigates subject mixing and subject neglect by editing the initial noise in three stages. First, DiDI performs **(1)** initial latent disentanglement (blue) to spatially isolate the subjects (e.g., cat-dog) into distinct regions of the latent. Concurrently, a **(2)** subject-aware partial denoising scheme (green) generates early semantic cues for each subject. From partially denoised latents, the top-$k$ features (top-5 in this case) are extracted and injected into disentangled latent through the **(3)** semantic feature injection module (red), as illustrated by the white patches. The resulting noise improves T2I alignment, even under poor initial noise.

attended by the token $y_i$. Hence, image regions with higher attention values share similar semantics with the token $y_i$. During the denoising process, these regions are reinforced through the injection of semantic features from $y_i$. Therefore, the content related to token $y_i$ tend to be synthesized at the regions with a high attention score.

While cross-attention maps interpret the attention from text tokens, the self-attention map $A_{\text{sa}}^{x,y} \in \mathbb{R}^{w/P \times h/P}$ models the relationships of a patch-$(x, y)$ with the rest of the patches. This is crucial for the generation of coherent structures in the final image.

**Attention Based Guidance.** Attend-and-Excite (Chefer et al., 2023) addresses subject neglect by introducing a loss that forces each subject token to attend to at least one spatial location with a high attention score. Specifically, it penalizes the weakest maximum attention score across the subject token set $\mathcal{Y}$ as:

$$\mathcal{L} = \max_{y \in \mathcal{Y}} \left(1 - \max_{x,y} \left(A_{\text{ca}}^{y}[x,y]\right)\right). \tag{3}$$

Self-Cross (Qiu et al., 2025) extends this concept to mitigate subject mixing. It aggregates a self-attention map from patches that are strongly attended by a subject, and penalizes its overlap with other subjects' cross-attention maps. For two subjects $y_1, y_2 \in \mathcal{Y}$, the loss is:

$$\sum_{x=1}^{w/P} \sum_{y=1}^{h/P} \min\left(A_{\text{as}}^{y_1}[x,y], A_{\text{ca}}^{y_2}[x,y]\right) + \sum_{x=1}^{w/P} \sum_{y=1}^{h/P} \min\left(A_{\text{ca}}^{y_1}[x,y], A_{\text{as}}^{y_2}[x,y]\right), \tag{4}$$

where $A_{\text{as}}^{y_1}$ is the aggregated self-attention map for $y_1$.

## 4 DiDI: DISENTANGLE DENOISE INJECT

Fig. 3 presents an overview of the proposed DiDI pipeline. This section details the design of its core components: initial latent disentanglement (Sec. 4.1), subject-aware partial denoising (Sec. 4.2), and semantic feature injection (Sec. 4.3).

(a) An example of subject mixing due to overlapping attention regions in the initial latent.

(b) An example of subject neglect caused by insufficient semantic features in the initial latent.

Figure 4: Cross-attention maps of SD failure cases. In both cases, failures in the final image can be traced back to the attention patterns in the early denoising timesteps.

## 4.1 INITIAL LATENT DISENTANGLEMENT

Recent studies (Marioriyad et al., 2024; Bao et al., 2024) identify attention overlap between subjects as a leading cause of subject mixing in diffusion models. While quantifying this overlap is challenging, cross-attention maps provide insight for analyzing and mitigating this issue. We aggregate the attention maps with a $(w/P \times h/P)$ resolution to obtain token-specific attention maps, $A_{\text{ca}} \in \mathbb{R}^{w/P \times h/P \times N}$. Following Chefer et al. (2023), we discard the attention to a $\langle SOT \rangle$ token and reweigh the scores for the remaining tokens using a Softmax operation to obtain the final attention maps.

Fig. 4 visualizes the cross-attention maps of subjects at different denoising stages. Notice the overlapping attention regions often emerge from early timesteps (see Fig. 4(a)). Notably, well-defined spatial layouts in the initial noise leads to distinct subject representations in the final image. In contrast, poorly separated subjects tend to merge during denoising, leading to subject mixing. Inspired by Marioriyad et al. (2024), DiDI leverages a Center of Mass (CoM) loss to penalize early attention overlap between subjects. The CoM loss can be formulated for two subject tokens $y_i$ and $y_j$ and their corresponding attention maps $A_{\text{ca}}^{y_i}$ and $A_{\text{ca}}^{y_j}$ as follows:

$$\mathcal{L}_{\text{CoM}}\left(A_{\text{ca}}^{y_i}, A_{\text{ca}}^{y_j}\right) = -\left\| \text{CoM}\left(A_{\text{ca}}^{y_i}\right) - \text{CoM}\left(A_{\text{ca}}^{y_j}\right) \right\|_2^2, \tag{5}$$

where $\text{CoM}\left(A_{\text{ca}}\right) = \left( \sum_{x=1}^{w/P} \sum_{y=1}^{h/P} x \cdot A_{\text{ca}}[x,y], \sum_{x=1}^{w/P} \sum_{y=1}^{h/P} y \cdot A_{\text{ca}}[x,y] \right)$. Specifically, this loss is applied over the initial latent $\mathbf{z}_T \sim \mathcal{N}(0,1)$, at the first denoising step to obtain the disentangled latent $\widetilde{\mathbf{z}}_T$. Using a predefined loss threshold is ineffective to accommodate diverse prompts, particularly when similar subjects require stronger disentanglement than others. Therefore, we apply the CoM loss for a fixed number of iterations in our experiments. Note that the CoM loss can be naturally extended to three or more subject synthesis tasks (see Appendix A.1.1).

## 4.2 SUBJECT-AWARE PARTIAL DENOISING

Vanilla SD often fails to generate multi-subject compositions. Still, it reliably generates individual subjects given the same noise seed. This suggests that the initial noise contains sufficient information to synthesize individual subjects, but lacks the semantics necessary to compose multi-subject scenes. The cross-attention layers in SD guide this composition by injecting token semantics into noise regions that already exhibit strong activations to synthesize subjects. However, when the initial latent lacks such regions i.e., semantics for a given subject, that subject is neglected. Hence, subject neglect can be attributed to the lack of its representative features in the initial latent as depicted in Fig. 4(b). Based on this insight, we inject the latent with characteristic semantics extracted from each subject token, while adhering to the learned SD distribution.

Naïve latent editing risks deviating from the standard Gaussian prior. Instead, DiDI employs a novel subject-aware partial denoising scheme. Given a shared initial latent $\mathbf{z}_T$, the input prompt is decomposed into constituent subjects and their attributes. For instance, the prompt "*a pink crown and a purple bow*" can be decomposed to "*pink crown*" and "*purple bow*". Then using a pretrained SD and the same noise schedule, DiDI generates partially denoised latents that are independently

conditioned on the subject embeddings as:

$$\mathbf{z}_t^{y_i} \leftarrow \text{SD}_{T \to t}(\mathbf{z}_T, y_i), \ \mathbf{z}_t^{y_j} \leftarrow \text{SD}_{T \to t}(\mathbf{z}_T, y_j). \tag{6}$$

Early denoising steps focus on perceptual compression i.e., removing high-frequency noise while learning semantic variation. Therefore, these latents retain proximity to the original noise while revealing early semantics of the corresponding subjects.

### 4.3 Semantic feature injection

Given the disentangled latent $\widetilde{\mathbf{z}}_T$ characterized by non-overlapping attention regions, and partially denoised latents $\mathbf{z}_t^{y_i}, \mathbf{z}_t^{y_j}$ encoding early semantic cues, DiDI enhances the subject semantics in $\widetilde{\mathbf{z}}_T$ by explicitly injecting representative features from $\mathbf{z}_t^{y_i}, \mathbf{z}_t^{y_j}$. For simplicity, we describe the procedure for a single subject; in multi-subject prompts, it is applied independently to each subject.

The cross-attention map of a subject token $y_i$ is used to select the top-$k$ patches with the highest attention scores. Specifically, $\mathcal{T}^{\text{D}} = \{\text{p}_1^{\text{D}}, \text{p}_2^{\text{D}}, \dots, \text{p}_k^{\text{D}}\}$ and $\mathcal{T}^{\text{L}} = \{\text{p}_1^{\text{L}}, \text{p}_2^{\text{L}}, \dots, \text{p}_k^{\text{L}}\}$ are the descending ordered top-$k$ patch sets for $\widetilde{\mathbf{z}}_T$ and $\mathbf{z}_t^{y_i}$, respectively. Patches with the same rank in $\mathcal{T}^{\text{D}}$ and $\mathcal{T}^{\text{L}}$ are then paired to form a swap set $\mathcal{S}$ as follows:

$$\mathcal{S} = \left\{ [\text{p}_i^{\text{D}}, \text{p}_i^{\text{L}}] \mid i \in \{1, \dots, k\} \right\}. \tag{7}$$

For each pair $[\text{p}_i^{\text{D}}, \text{p}_i^{\text{L}}] \in \mathcal{S}$, the patch $\text{p}_i^{\text{D}}$ is replaced to its counterpart from the subject-specific latent $\text{p}_i^{\text{L}}$. Note that each replacement modifies a $(P \times P)$ region in the disentangled latent. Fig. 3 shows the process for a two subject composition, which can be extended to multiple concepts (see Fig. 12). Unlike previous methods (Mao et al., 2023) that relocate features within a single latent for layout control, our approach, inspired by the cross-attention mechanism, injects semantically enriched regions from deliberately crafted external latents into a spatially disentangled latent. The modified noise $\hat{\mathbf{z}}_T$ is then used for the final image synthesis via the standard SD denoising, or together with other optimization strategies.

The overall pipeline for DiDI (see Alg. 1 in Appendix A.1.2) integrates the self-cross guidance Eqns. (3, 4) to guide the first half of the denoising process beyond the initial latent.

## 5 Experiments

**Implementations and Baselines.** This study uses the standard SD v1.4[1] T2I model as the experimental baseline. All results are generated using a fixed guidance scale of 7.5 and 50 inference steps ($T = 50$). Following Chefer et al. (2023), we apply a Gaussian smoothing function ($3 \times 3$ kernel, $\sigma = 0.5$) for the cross-attention maps. The target tokens are identified according to the total token length of a prompt. For DiDI, the CoM loss is applied for 15 iterations, partial denoising is carried out for the first 10 timesteps, and $k = 8$ patches are used for injection in all experiments. Additional implementation details are provided in Appendix A.1.3.

Our study compares DiDI with another initial noise optimization method InitNO (Guo et al., 2024) on three training-free baselines: (1) vanilla SD, (2) Attend-and-Excite (Chefer et al., 2023), and (3) Self-Cross (Qiu et al., 2025). We find that our method consistently outperforms InitNO across different compositions.

**Datasets.** We adopt three datasets (Chefer et al., 2023) that target common failure modes in SD: Animal-Animal, Animal-Object, and Object-Object compositions. We also report results on the Similar Subjects Dataset (SSD-2) (Qiu et al., 2025) which involves semantically related but visually distinct subjects (e.g., duck and penguin), as a challenging benchmark to evaluate subject mixing. While the experiments are focused on these datasets, DiDI can be easily extended to more complex and diverse prompts (see Appendix A.2.2).

### 5.1 Qualitative analysis

Fig. 5 presents a qualitative comparison of ours against the baseline SD v1.4, InitNO (Guo et al., 2024), and Self-Cross (Qiu et al., 2025) using identical prompts and same set of random seeds[2].

---

[1]Available at: https://huggingface.co/CompVis/stable-diffusion-v1-4

[2]NSFW content is returned as a black image, and discarded for quantitative analysis.

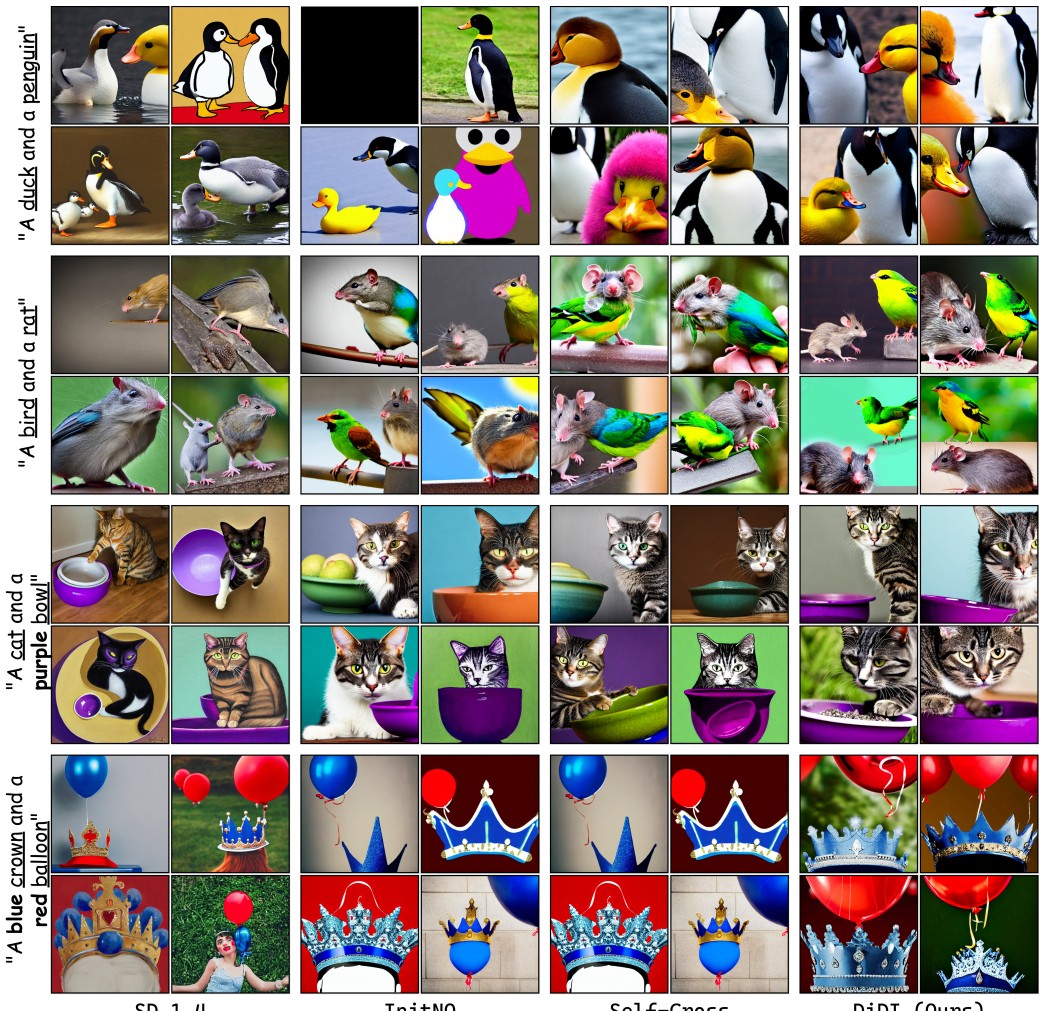

Figure 5: Qualitative comparison of DiDI (ours), SD v1.4, InitNO (Guo et al., 2024), and Self-Cross (Qiu et al., 2025). For each prompt, four images per method are generated using the same set of random seeds. The subject tokens are underlined, and attributes used by DiDI are shown in **bold**.

As seen in the first two prompts, existing methods, continue to be vulnerable to subject mixing, specially when dealing with semantically similar subjects. For instance, InitNO and Self-Cross often generate fused subjects for the prompt "*a bird and a rat*". In contrast, our method which explicitly disentangles subjects and promotes individual generation, renders both entities realistically with their characteristic semantics.

The bottom two prompts highlight the challenge of attribute binding. Both InitNO and Self-Cross often misplace attributes like colors, e.g., binding "*purple*" to the background instead of the bowl, or confusing the color assignments between the balloon and crown. By injecting subject-specific semantics from partially denoised latents, our approach consistently achieves accurate bindings, leading to more faithful generations.

Furthermore, the baseline methods tend to suffer from subject neglect and produce cartoonish figures (see third row, Fig. 11), whereas our method preserves subject fidelity across most settings. Particularly, InitNO and Self-Cross demonstrate sub-optimal optimization in certain categories (e.g., Object–Object prompts), likely due to shared initial latents; this also leads to similar outputs. Additional qualitative results, including direct comparisons between InitNO and our method without test-time guidance, are provided in Appendix A.2.2.

Table 1: Quantitative comparison using averaged TIFA and CLIP Text-Text similarities. Results for the Animal-Animal (A-A), Animal-Object (A-O), Object-Object (O-O), and SSD-2 datasets, are reported using the same set of 64 random seeds for each method.

| Model | TIFA (↑) | | | | CLIP Text-Text (↑) | | | |
|---|---|---|---|---|---|---|---|---|
| | A-A | A-O | O-O | SSD-2 | A-A | A-O | O-O | SSD-2 |
| Stable Diffusion 1.4 | 63.93 | 79.67 | 60.56 | 49.47 | 76.5 | 79.2 | 76.4 | 70.0 |
| + InitNO | 70.70 | 87.97 | 70.77 | 48.95 | 80.9 | 82.4 | **80.3** | 71.6 |
| + DiDI (ours) | **72.05** | **89.94** | **73.36** | **50.05** | **81.7** | **83.5** | 79.9 | **72.0** |
| Attend-and-Excite | 72.70 | 91.32 | 75.39 | 50.30 | 80.6 | 83.0 | 81.1 | 70.8 |
| + InitNO | 74.83 | 92.36 | **78.46** | 50.70 | 82.2 | 84.1 | **82.3** | 72.0 |
| + DiDI (ours) | **76.32** | **92.42** | 70.58 | **51.58** | **83.2** | **84.2** | 75.6 | **72.8** |
| Self-Cross | 77.85 | 92.83 | 80.71 | 52.55 | 83.2 | 84.6 | 82.0 | 72.3 |
| + InitNO | 77.79 | 93.14 | 78.20 | 52.48 | 84.2 | **84.7** | **82.3** | 73.4 |
| + DiDI (ours) | **78.85** | **93.34** | **82.43** | **52.87** | **84.8** | **84.7** | 81.4 | **74.0** |

## 5.2 QUANTITATIVE ANALYSIS

The T2I faithfulness can be evaluated using, TIFA-VQA (Hu et al., 2023) scores for fine-grained assessment of multi-subject compositions, and CLIP text-text similarity scores for global T2I alignment. A fixed set of 64 random seeds is used to generate images for each prompt, across all methods.

**TIFA-VQA scores.** TIFA evaluates the faithfulness of generated images to their prompt by decomposing them into semantic elements like subjects, colors, and other attributes. Our implementation uses open-source models: Llama-2 (Touvron et al., 2023) as the LLM and BLIP (Li et al., 2022) as the VLM. More implementation details can be found under Appendix A.1.4.

As shown in Table 1, the proposed method consistently outperforms prior SOTA across all datasets. Overall, the largest gain is observed in the Object-Object category, where our method achieves 82.43%, improving by a significant 4.2% compared to the previous best. On the challenging SSD-2 dataset, DiDI surpasses the current benchmark by 0.4%. This performance also generalizes to simpler and mixed compositions, as reflected by the Animal-Animal and Animal-Object results. These results underscore DiDI's ability to synthesize faithful images across diverse settings.

**CLIP Text-Text Similarity.** Following Chefer et al. (2023), we generate BLIP captions for each of the 64 images per prompt and compute their CLIP similarity with the original prompt. This is repeated and averaged between all prompts to obtain the final score. The results in Table 1 show that our method surpasses the existing SOTA on SSD-2 and Animal-Animal, while achieving similar performance on the Animal-Object set. However, the performance falls within 1% of the best in the Object-Object category. This is attributed to CLIP's bias toward dominant features (Abbasi et al., 2025) and its limited ability to capture fine-grained attributes (e.g., colors), making it less reliable for Object-Object prompts. Sec. A.2.3 provides additional discussion on these biases.

## 5.3 ABLATION STUDY

We conduct our ablation study on a randomly sampled subset of 30 prompts, evenly drawn from the Animal-Animal, Animal-Object, and Object-Object datasets.

**Impact of CoM loss.** The CoM loss is introduced to mitigate subject mixing in the initial latent. Ablating it from the pipeline results in a 0.82% drop in TIFA score and a 0.60% drop in text-text similarity (see Fig. 6). Qualitatively, this causes subject overlap, where semantics from the "*bear*" token erroneously attend to regions of the "*rabbit*". These findings highlight the role of CoM loss in preserving subject localization and preventing attention overlap.

**Impact of feature injection.** Semantic feature injection is designed to reinforce the subject semantics in the initial latent; ideally, improving both subject fidelity and attribute binding. Removing this module causes a 0.56% drop in TIFA score and a 0.03% drop in text-text similarity. Qualitatively, as shown in Fig. 6, this ablation leads to incorrect color binding ("*gray suitcase*" → red), and subject

| Component | TIFA (↑) | Text-Text (↑) |
|---|---|---|
| DiDI (ours) | 85.52 | 84.04 |
| - Lcom | 84.70 | 83.44 |
| - injection | 84.96 | 84.01 |
| - denoising | 85.01 | 84.02 |
| SD v1.4 | 70.24 | 78.03 |

Figure 6: Ablation study on the components of DiDI: $\mathcal{L}_{\mathrm{CoM}}$, semantic feature injection, and subject-aware partial denoising. Ablating $\mathcal{L}_{\mathrm{CoM}}$ leads to subject mixing, while injection improves subject fidelity and attribute binding.

neglect (*"rabbit with a glasses"* → only glasses). In comparison, the full pipeline faithfully renders all entities. These results highlights the role of feature injection in improving T2I alignment.

**Impact of partial denoising.** We evaluate the role of subject-aware partially denoised latents as sources for feature injection by comparing them against InitNO-optimized latents. Replacing these latents with InitNO variants causes a 0.51% drop in TIFA score and a 0.02% decrease in text-text similarity. Qualitatively, this substitution reduces subject fidelity (e.g., cartoonish rabbit) and generates incorrect attribute bindings (e.g., red suitcase). These results emphasize the importance of our partial denoising scheme to preserve subject semantics in high quality injections.

# 6 CONCLUSION

This study investigated how the initial noise affects compositional failures in T2I generation such as subject mixing, subject neglect, and incorrect attribute binding. Our findings reveal that these issues arise in early denoising stages due to entangled subject representations and insufficient semantic information. To address this, we proposed DiDI, a training-free pipeline that optimizes the initial noise for faithful T2I synthesis. DiDI consists of three core processes: (1) initial latent spatial disentanglement, (2) subject-aware partial denoising, and (3) semantic feature injection, designed to address subject mixing and subject neglect. Unlike existing approaches, DiDI directly edits the latent, improving semantic alignment and subject fidelity for both good and bad seeds. Furthermore, the subject-aware partial denoising scheme helps overcome the limitations of text encoders in multi-subject settings.

Future work will explore the extension of the DiDI framework to broader generative tasks beyond T2I synthesis. Studies (Abbasi et al., 2025) suggest that subject token attention might follow a hierarchy based on the order of appearance in the prompt; adapting DiDI to dynamically account for such patterns is a promising direction. In addition, rather than relying on feature injection from a single timestep, leveraging information across multiple timesteps could further enhance generation fidelity and finer-grained control.

REPRODUCIBILITY STATEMENT

We implement our method and all baselines using PyTorch (Paszke et al., 2019) and the Hugging Face Transformers library (Wolf et al., 2019). Implementation details are provided in Sec. 5 and A.1.3. All experiments were conducted on open-source benchmarks, without dependence on external APIs. Upon acceptance, the complete source code, including the full set of seeds and detailed reproduction instructions will be released for the research community to fully access, reproduce, and extend our work.

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

# A APPENDIX

## A.1 ADDITIONAL DETAILS

### A.1.1 EXTENDED LOSS FUNCTION

For prompts containing three or more subjects, $\mathcal{L}_{\text{CoM}}$ (Eq. 5) can be extended by computing the area of the polygon formed by CoMs of each subject's cross-attention map.

$$\mathcal{L}_{\text{CoM}}\left(A_{\text{ca}}^1, \ldots, A_{\text{ca}}^N\right) = -\text{Area}(\mathcal{P}), \tag{8}$$

where $\mathcal{P}$ is the polygon formed by the CoMs of $N$ subjects.

### A.1.2 DiDI ALGORITHM

Alg. 1 outlines the DiDI pipeline. Our method operates exclusively at the first denoising timestep, and is compatible with both initial noise optimization methods (Guo et al., 2024) and test-time optimization methods including (Chefer et al., 2023; Qiu et al., 2025), for enhanced performance.

---

**Algorithm 1:** DiDI

---

**Input:** Pre-trained T2I diffusion model $\text{SD}(\cdot)$,
text prompt $y$ and subject token set $\mathcal{Y}$,
Iteration threshold: $\tau_{\text{COM}}, \tau_{\text{steps}}$.
**Result:** Generated image $\mathbf{x}$.
Initialize $\mathbf{z}_T \sim \mathcal{N}(0,1), t \leftarrow \text{T}$
$\_, A_{\text{ca}} \leftarrow \text{SD}\left(\mathbf{z}_T, y, t\right)$
**for** $i = 1$ **to** $\tau_{\text{COM}}$ **do**
    Calculate $\mathcal{L}_{\text{CoM}}$ (Eq. 5)            ▷ Latent disentangling
    $\widetilde{\mathbf{z}}_T \leftarrow \text{SGD}\left(\mathcal{L}_{\text{CoM}}\right)$
    $\_, A_{\text{ca}} \leftarrow \text{SD}\left(\widetilde{\mathbf{z}}_T, y, t\right)$
**end**
**for** $y_i \in \mathcal{Y}$ **do**
    $\mathbf{z}_t^{y_i} \leftarrow \mathbf{z}_T$            ▷ Partial denoising
    **for** $t = 1$ **to** $\tau_{\text{steps}}$ **do**
        $\mathbf{z}_t^{y_i} \leftarrow \text{SD}\left(\mathbf{z}_t^{y_i}, y_i, t\right)$ (Eq. 6)
    **end**
    $\mathcal{T}^{\text{L}} \leftarrow \text{Top-k}\left(\mathbf{z}_t^{y_i}, y_i, k\right), \mathcal{T}^{\text{D}} \leftarrow \text{Top-k}\left(\widetilde{\mathbf{z}}_T, y_i, k\right)$
    **for** $p_i^{\text{L}} \in \mathcal{T}^{\text{L}}, p_i^{\text{D}} \in \mathcal{T}^{\text{D}}$ **do**
        $\widetilde{\mathbf{z}}_T(p_i^{\text{D}}) \leftarrow \mathbf{z}_t^{y_i}(p_i^{\text{L}})$ (Eq. 7)      ▷ Swap patches
    **end**
**end**
$\hat{\mathbf{z}}_T \leftarrow \widetilde{\mathbf{z}}_T$
$\mathbf{x}, \_, \leftarrow \text{SD}\left(\hat{\mathbf{z}}_T, y, t\right)$ or $\text{Self-Cross}\left(\hat{\mathbf{z}}_T, y, t\right)$     ▷ Denoise
**return** $\mathbf{x}$

---

### A.1.3 IMPLEMENTATION DETAILS

We adopt the official SD v1.4 T2I model with the CLIP ViT-L/14 text encoder in our study. Following prior work (Chefer et al., 2023), we extract the cross-attention maps at a $16 \times 16$ resolution with a patch size $P = 4$, followed by a Gaussian smoothing with a $3 \times 3$ kernel ($\sigma = 0.5$). We also apply classifier-free guidance (Ho & Salimans, 2022) using a scale of 7.5 for a total of 50 ($T = 50$) inference steps. The CoM loss is applied iteratively for 15 steps. For partial denoising, we use the vanilla SD v1.4 model, independently denoising each subject for the first 10 timesteps ($t = 10$). A shared initial latent is used in both, the main and per-subject denoising to prevent any out-of-distribution artifacts. During semantic feature injection, we select and use the top-8 ($k = 8$) patches. After DiDI, we apply Self-Cross guidance (Qiu et al., 2025) during the first half (25 steps) of inference. All experiments were conducted on a single NVIDIA RTX 4090 (24GB) GPU.

We evaluate DiDI on public datasets where each prompt includes a conjunction of two subject tokens (see Table 2). During prompt decomposition, we extract each noun along with its preceding token to

Table 2: Items composing the Animal-Animal, Animal-Object, Object-Object, and SSD-2 datasets. The table also reports the number of prompts and the total number of generated images per dataset.

| Category | Items |
|---|---|
| Animals | bird, bear, cat, dog, elephant, frog, horse, lion, monkey, rabbit, rat, turtle |
| Objects | apple, backpack, balloon, bench, bow, bowl, car, chair, clock, crown, glasses, suitcase |
| Colors | black, blue, brown, gray, green, orange, pink, purple, red, white, yellow |
| Similar Subjects | aster, beagle, bus, carnation, catfish, cheetah, chihuahua, collie, condor, corgi, cow, crab, dahlia, donkey, dolphin, eagle, fly, horse, hummingbird, husky, jeep, kingfisher, leopard, loon, manatee, marlin, mastiff, orca, owl, ox, panther, parrot, peasant, penguin, pheasant, pigeon, poodle, pug, rose, seal, sedan, sparrow, spider, sunflower, swan, tiger, truck, trout, woodpecker, zebra |

| Dataset | No. prompts / No. Images |
|---|---|
| Animal-Animal | 66 / 4224 |
| Animal-Object | 144 / 9216 |
| Object-Object | 66 / 4224 |
| SSD-2 | 31 / 1984 |

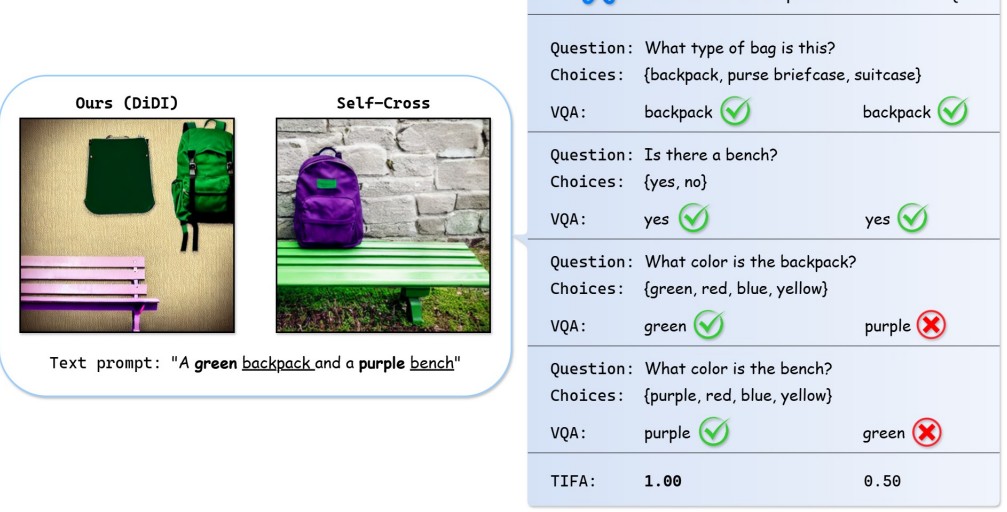

Figure 7: Example of TIFA score computation illustrated using two images generated by DiDI (left) and Self-Cross (right) using the same prompt and seed. Llama-2 generates multiple-choice questions based on the prompt, and BLIP answers them using the image. The final TIFA score is the average across all questions.

get subject spans. For instance, the prompt "*A cat and a dog*" becomes "*A cat*" and "*A dog*", while "*A yellow bow and a pink bowl*" becomes "*Yellow bow*" and "*Pink bowl*". Although this is tailored to the datasets used in our study, the pipeline is flexible and can be easily extended to support other prompt formats (see Fig. 12).

### A.1.4 TIFA VQA

We use TIFA-VQA to assess fine-grained T2I alignment in concepts such as objects, animals, and colors, which closely align with our datasets. For each prompt, an LLM (LLaMA-2) generates multiple-choice questions and reference answers based on the detected concepts. These questions

are answered by a VQA model (BLIP), using the generated image. Fig. 7 shows an example comparing an output from Self-Cross and DiDI on an Object-Object prompt. For clarity, this example has been simplified; in practice, the number of questions varies with the concepts present in the prompt. Overall, TIFA provides an accurate evaluation of T2I faithfulness.

Unlike the standard TIFA setup (Hu et al., 2023), we remove the question filtering step, which is responsible for filtering redundant questions (e.g., multiple queries about the same subject). We observed that this filtering can also discard complimentary questions that capture different aspects. For instance, the questions:

1. Is there a suitcase?

   Choices: {yes, no}

2. What type of luggage is this?

   Choices: {suitcase, backpack, briefcase, purse}

both target the same object but evaluate different failures: subject neglect and subject mixing. Therefore, we retain all LLM generated questions without filtering. This change is applied to all methods and does not introduce bias, since the worst-case is redundancy.

## A.2 ADDITIONAL EXPERIMENTS

### A.2.1 VARIATIONS OF SEMANTIC FEATURE INJECTION

Fig. 8 illustrates the effect of varying semantic feature injection parameters, specifically, the timestep of extracting partially denoised latents ($t$) and the number of injection patches ($k$). Early timesteps ($t = 5$) lack sufficient semantic cues, leading to weak feature transfer and incomplete subject generation. Similarly, using too few patches ($k = 4$) leads to incomplete feature transfer, often causing subject neglect. This trade-off can be somewhat mitigated by balancing the parameters, i.e., using higher timesteps with fewer patches, or lower timesteps with more patches. However, higher $t$ or $k$ values risk introducing artifacts by pushing the latent to out-of-distribution or by injecting redundant semantics. We find that moderate settings ($t = 10, k = 8$) strike the best balance. Notably, the number of patches play a more significant role compared to the timestep (see last row), as with higher patch $k$ the model still generates both subjects even at lower timesteps despite synthesizing random objects sharing similar attributes. The latter is due to excessive information.

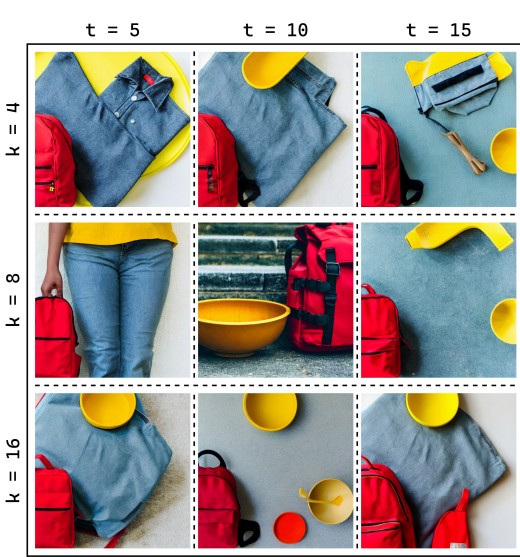

Figure 8: Visualizing the impact of different parameters in semantic feature injection.

Table 3: Averaged Image Reward scores, reported over 64 seeds.

| Model | Image Reward (↑) | | | |
| --- | --- | --- | --- | --- |
| | A-A | A-O | O-O | SSD-2 |
| Stable Diffusion 1.4 | -0.11 | 0.40 | -0.59 | -0.14 |
| + DiDI (ours) | **0.74** | **1.31** | **0.48** | **0.00** |
| Self-Cross | 1.32 | **1.46** | 1.27 | **0.35** |
| + DiDI (ours) | **1.35** | 1.44 | **1.30** | 0.34 |

### A.2.2 Qualitative Experiments

**Image Reward Scores.** In addition to TIFA, we evaluate the generations of DiDI using ImageReward: a metric for evaluating human preference in T2I generation. ImageReward outperforms alternatives such as the Aesthetic score. Other metrics like FID rely on a reference dataset and averaging over the entire dataset, which limits their reliability under diverse and complex prompts. As shown in Table 3, integrating DiDI into the vanilla SD leads to consistent improvements across all datasets. When combined with Self-Cross, we observe competitive performance with a modest gain of 0.03% in the Animal-Animal and Object-Object compositions. This reduced effect is expected, as DiDI disentangles the subjects at the initial latent while Self-Cross predominantly focuses on subject mixing. These overlapping processes lead to occasional artifacts like concatenated images (see Sec. A.2.4). Overall, these results demonstrate that DiDI enhances the initial latent noise to achieve more faithful generations while improving or preserving the image fidelity.

**Additional Qualitative Results.** Additional qualitative comparisons are provided in Fig. 10, 11, and 12. Fig. 10 compares DiDI with InitNO solely under initial noise optimization, excluding any test-time optimization. Fig. 11 compares DiDI with SD v1.4, InitNO, and Self-Cross on two-subject prompts. Fig. 12 shows the extension of DiDI to more complex two subject and three subject cases. As shown, DiDI synthesizes faithful outputs, even with the base SD v1.4 model, and scales effectively to more complex settings.

### A.2.3 Biases of CLIP Text-Text Similarity

DiDI achieves SOTA results in the Object-Object dataset on the TIFA benchmark, yet shows a small drop in CLIP text-text similarity, still remaining within 1% of the best model i.e., Self-Cross. Despite this, it outperforms the vanilla SD across all categories. We find that CLIP similarity, while effective under most, is unreliable for evaluating the faithfulness in Object-Object compositions. Specifically, CLIP often produce high scores when the correct subjects are present, even with incorrect attributes that are misaligned with the prompt. Fig. 9 illustrates a few examples in which our method (on the left) generates more faithful images than Self-Cross (on the right) using the same prompts and random seeds. However, for all these images, the CLIP score assigned to ours is significantly lower compared to Self-Cross. As pointed out in prior studies (Abbasi et al., 2025; Kang et al., 2025), this behavior

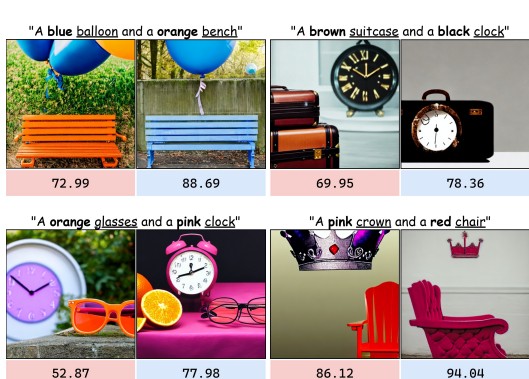

Figure 9: CLIP text–text similarity scores (↑) for images generated with DiDI (left) and Self-Cross (right). While it captures high-level alignment for salient subjects, these scores overlook finer attributes making them less reliable for evaluating faithfulness in Object–Object settings.

comes from CLIP's bias toward salient features in the image and its limited ability to capture more fine-grained attributes such as colors which are present in the Object-Object class. To conclude, our study relies on the more suitable TIFA benchmark to evaluate the T2I faithfulness in the Object–Object dataset.

### A.2.4 Limitations of DiDI

DiDI operates only at the initial noise stage, and is combined with existing test-time methods to optimize the consequent denoising process. While DiDI mitigates issues like subject mixing and neglect, and improves subject fidelity to produce realistic images, it can still produce unsatisfactory results, including concatenated or blurry images. Similar failure cases are also observed in InitNO and Self-Cross. Some representative cases are shown in Fig. 13. We assume that concatenated images (see top row Fig. 13) can be alleviated using a dynamic CoM loss or layout guidance paired with an appropriate test-time optimization method. To address blurriness (see bottom row Fig. 13), future work could explore the aggregation of attention maps at multiple resolutions, and their coordination between several immediate timesteps to preserve the finer details.

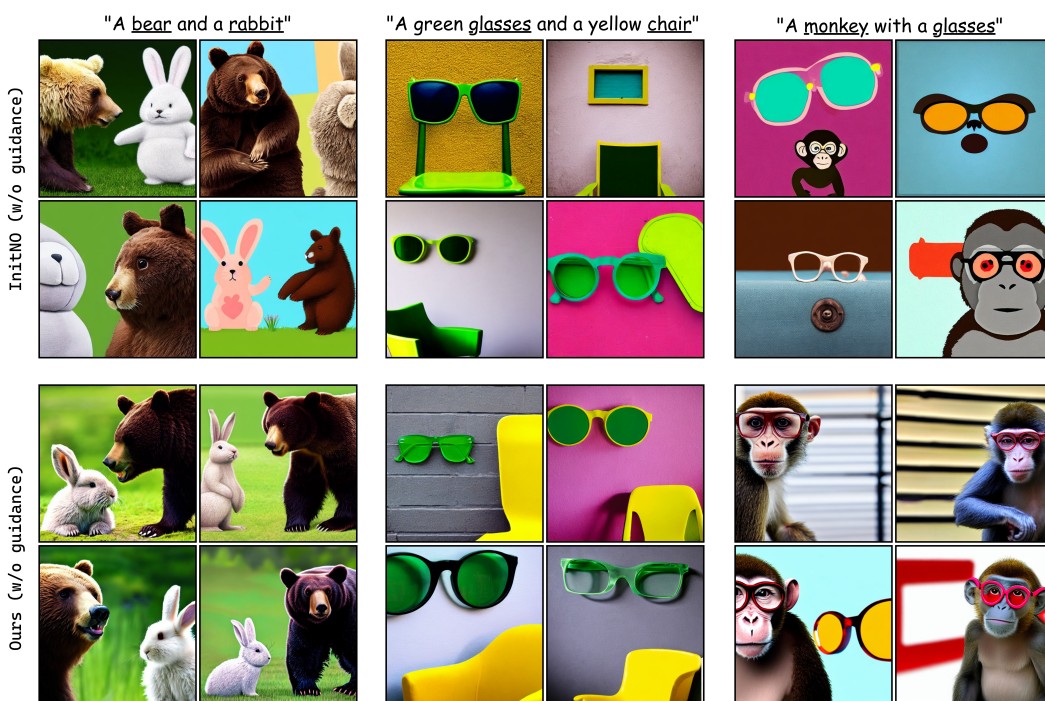

Figure 10: Additional qualitative comparison between DiDI (ours) and InitNO (Guo et al., 2024) without test-time guidance. This comparison highlights the effects of the initial latent. For each prompt, we show four images per method generated by the same set of random seeds. The subject tokens considered by all methods are underlined, and the attributes used by DiDI are shown in **bold**. DiDI produces more faithful and higher-fidelity generations compared to InitNO.

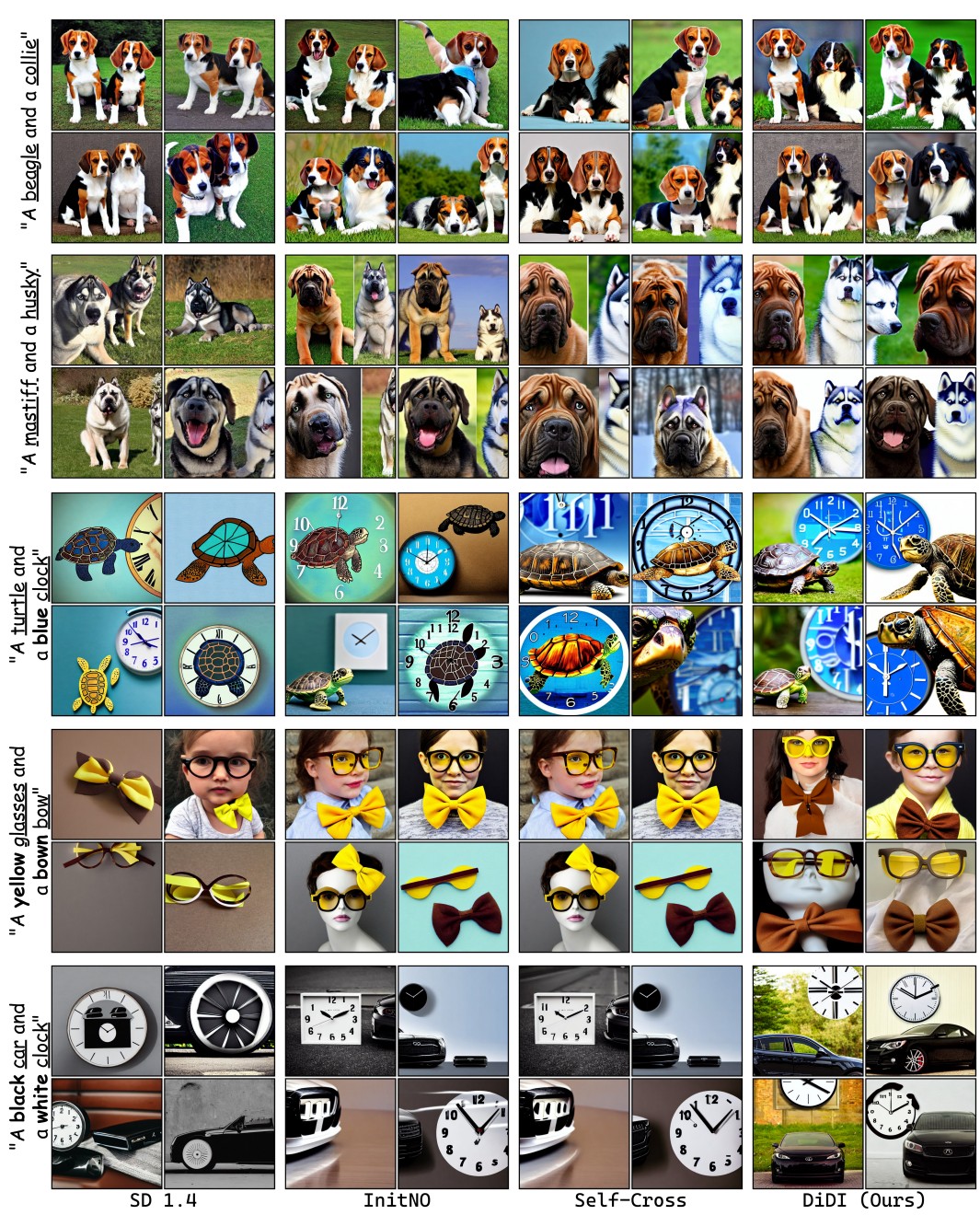

Figure 11: Qualitative comparison between DiDI (ours), SD v1.4, InitNO (Guo et al., 2024), and Self-Cross (Qiu et al., 2025). For each prompt, we show four images per method generated by the same set of random seeds. The subject tokens considered by all methods are underlined, and the attributes used by DiDI are shown in **bold**.

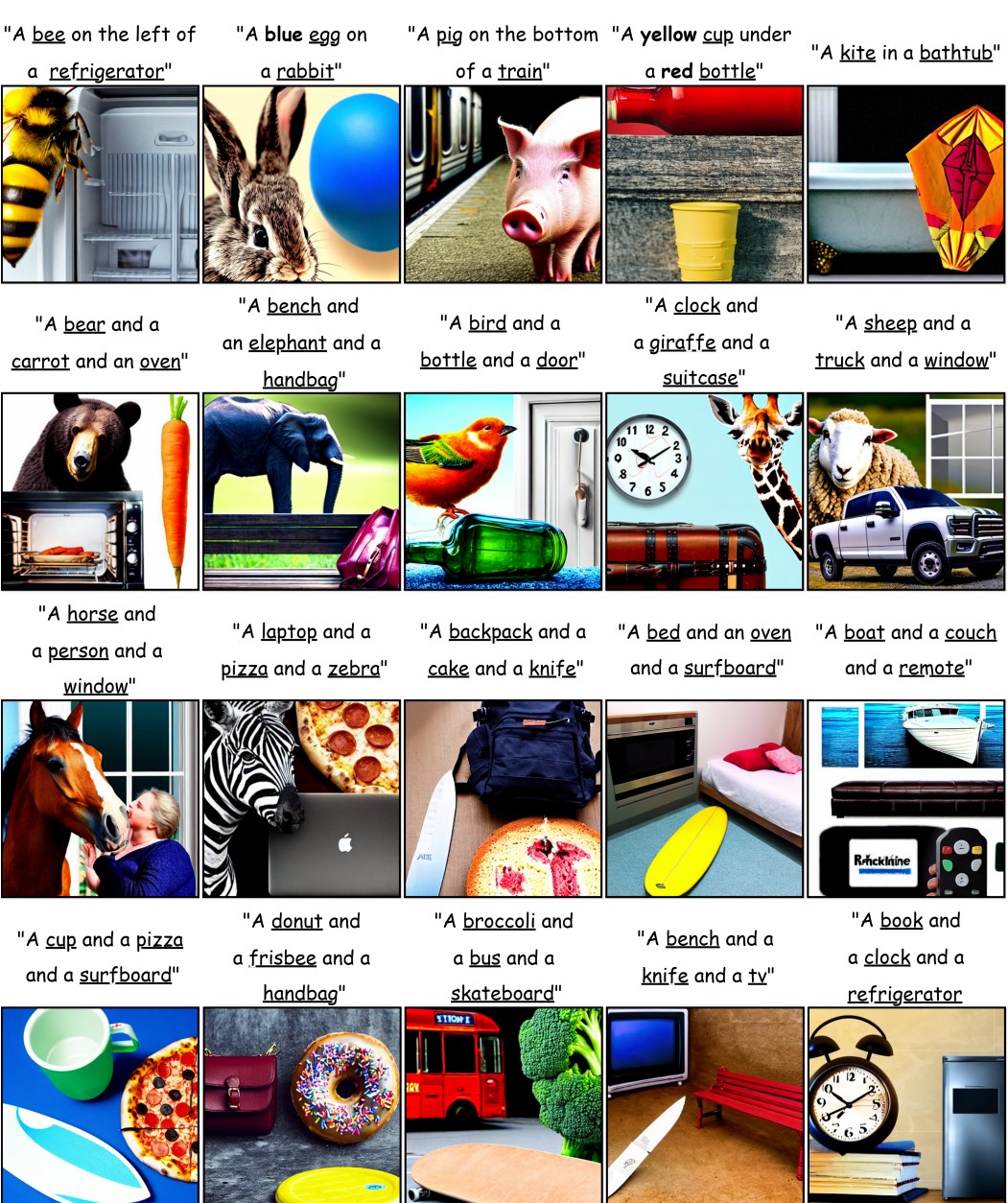

Figure 12: Qualitative results from DiDI on three-subject prompts. These examples demonstrate that DiDI scales effectively to more complex scenarios. The subject tokens considered by DiDI are underlined, and the attributes used by DiDI are shown in **bold**.

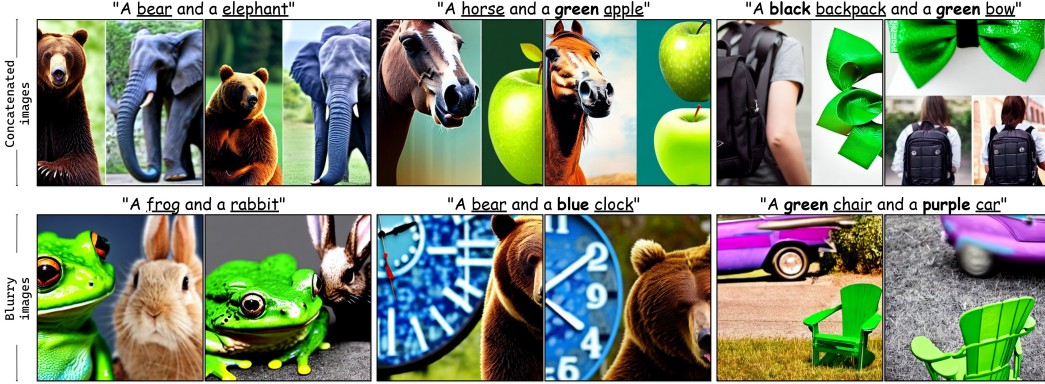

Figure 13: Representative failure cases of DiDI. DiDI occasionally produces (1) concatenated outputs or (2) blurry generations. The subject tokens considered by DiDI are underlined, and the attributes used by DiDI are shown in **bold**.

