# OpenReview forum: "DiDI: Disentangle Denoise Inject for Improving T2I Diffusion Models"
_ICLR.cc/2026/Conference — ICLR 2026 Conference Withdrawn Submission_

### Official Review · Reviewer_DtEW · 2025-10-15

**Soundness:** 2
**Presentation:** 3
**Contribution:** 2
**Rating:** 2
**Confidence:** 5

**Summary:**

This paper tackles the problem of subject mixing and attribute binding in SD1.4. It proposes a method to disentangle the objects from the cross-attention maps. It also introduces a method to inject independently updated latents into the composed image. The experiments show advantages over previous methods on mitigating subject mixing and improving attribute binding.

**Strengths:**

1. This paper is well presented, with clear figures, ablation effects, and qualitative results.
2. This paper's evaluations and benchmarks reasonably reflect the problem that this paper mainly tackles and shows the strengths of the proposed approaches over other baselines.
3. The proposed method is orthogonal to combine with other approaches such as InitNO and Self-Cross.

**Weaknesses:**

1. The experiments are limited to SD1.4. It is not clear how the proposed approaches can **generalize to other UNet-based models**, such as SD2.1 and SDXL.
2. This paper mainly designs methods for UNet-based models, which are no longer prevalent in terms of ICLR2026. The authors should conduct **studies and experiments on more recent DiT-based models**, including SD3.5, FLUX, Qwen-Image, etc. Noticeably, since subject mixing issues have been much less visible on latest models, it questions the **significance of the problem** that this paper is dealing with.

**Questions:**

1. In 4.2 and 4.3, the authors introduce "subject-aware partial denoising" which updates the latents independently. **Would this break the spatial relationships and interactions of objects?** For example, some examples in Fig.12 show wrong relationships: a blue egg on a rabbit, a kite in a bathtub, a pig on the bottom of a train. Some quantitative numbers will be more convincing.
2. **How efficient is the proposed method, in terms of the amount of time to generate N successful samples?** From the current benchmarks, I can see the proposed method has a higher win rate. But it also comes with increasing computations. Original SD may be able to generate the same amount of correct samples with more runs, while each run takes less time. To evalute this, one can record FLOPs or latency to successfully generate N samples.

---

### Official Review · Reviewer_X83U · 2025-10-28

**Soundness:** 3
**Presentation:** 3
**Contribution:** 2
**Rating:** 4
**Confidence:** 4

**Summary:**

The authors propose a training-free method, DiDI, built upon Stable Diffusion, to enhance text–image alignment in diffusion-based T2I generation. The core idea involves editing the initial latent noise and refining the denoising process to improve compositional consistency. The proposed method is implemented on SD 1.4 and shows superior performance compared to several baselines such as Attend-and-Excite.

**Strengths:**

1. The paper is overall well-structured and complete.
2. The methodology is clearly described, with a coherent explanation of each module’s role and function.

**Weaknesses:**

1. The proposed method is built on Stable Diffusion 1.4, which is a relatively weak base model.
   Since the method is plug-and-play, it should be evaluated on more base models to demonstrate its generalization ability.Moreover, state-of-the-art T2I models (e.g., those using the MMDiT architecture) are significantly more capable than SD1.4. It remains unclear whether DiDI is still effective when applied to such newer models.Given that SD1.4 is an early-generation model and modern backbones, such as Flux, SD3.5, Qwen image, and so on, already exhibit much stronger compositional capabilities, the motivation of the study feels insufficient. The authors should better clarify the relevance and impact of this work in today’s context.
2. The baselines are not sufficiently comprehensive. Several recent training-free alignment methods (e.g., CONFORM[1], Divide & Bind[2], ToMe[3]) were not included in the comparison, which limits the strength of the experimental validation.
3. The core idea of DiDI is conceptually similar to previous training-free refinement methods. Although the authors claim that using the CoM to edit the initial noise is different from prior methods, it is highly reminiscent of approaches like Attend-and-Excite[4] and Divide & Bind[2], which also refine the latent at the first denoising step.

[1]: conform: contrast is all you need for high-fidelity text-to-image diffusion models

[2]: Divide & bind your attention for improved generative semantic nursing

[3]: Token Merging for Training-Free Semantic Binding in Text-to-Image Synthesis

[4]: Attend-and-Excite: Attention-Based Semantic Guidance for Text-to-Image Diffusion Models

**Questions:**

1. What is the motivation for conducting this study based on SD1.4 rather than a more recent backbone?
2. What are the fundamental differences between DiDI and prior methods such as Attend-and-Excite or Divide & Bind?
3. The generated image quality appears degraded in some examples, can the authors explain why?

**Details Of Ethics Concerns:**

I have no ethical concerns regarding this paper.

---

### Official Review · Reviewer_FChj · 2025-10-31

**Soundness:** 2
**Presentation:** 3
**Contribution:** 2
**Rating:** 4
**Confidence:** 4

**Summary:**

This paper introduces DiDI, which is a training-free pipeline designed to mitigate image-text alignment issues in T2I diffusion models when handling multi-subject prompts. The authors propose that compositional failures fundamentally stem from poor initial noise, leading to entangled subject representations and insufficient semantics in early denoising stages. DiDI addresses this through: (1) Initial Latent Spatial Disentanglement using a COM loss to separate subjects spatially, (2) Subject-Aware Partial Denoising of individual subject prompts to generate early semantic cues, and (3) Semantic Feature Injection via patch swapping to integrate these semantic cues into the spatially optimized latent.

**Strengths:**

1. The key intuition that when these models are used to generate individual objects, they usually work perfectly well- and hence leveraging that while trying to generate images with multiple objects is both intuitive and resonates well
2. Idea of using center of mass compared to majority prior works directly trying to operate on holistic attention maps seems interesting

**Weaknesses:**

1. The qualitative results are not that appealing and seem to be a bit unnatural in many scenarios. For instance, in figure 5, all the results generated by DiDI seem to be very zoomed in and most objects are not entirely visible e.g. the balloon in the last row.
2. The COM loss would potentially hinder scenarios where there is a desirable spatial overlap between objects.. e.g. for a prompt like "a turtle inside a bowl", there would be a desirable/required overlap between the subjects to generate such an image. This aspects seems to limit the practical utility of the proposed solution.
3. The results shown in the paper consider only simpler 2-subject prompt scenarios, and not much results are included on the real challenging scenarios with multiple interacting subjects
4. Stable Diffusion 1.4 is the only model authors have compared with, which is limiting given that the applicability of the proposed method on most of the latest models is questionable.
5. The main results in Table 1 do not compare with many prominent and important baselines in this field (mentioned below) which have tackled the same challenging of improving image-text alignment for diffusion models in multi-subject scenarios/incorrect attribute binding. The authors have compared with only a single baseline InitNO which seems insufficient.

Some baselines to consider:

[a] Attend-and-excite: Attention-based semantic guidance for text-to-image diffusion models (ACM TOG 2023)

[b] Test-time attention segregation and retention for text-to-image synthesis (ICCV 2023)

[c] Conform: Contrast is all you need for high-fidelity text-to-image diffusion models (CVPR 2024)

**Questions:**

1. Wouldn't the patch swapping be affected by the relative size in which the model might try to generate a dog when generating a dog alone vs when generating a dog with 2 more objects?
2. Given the attention maps obtained with partial denoising would be noisy, how accurate are the computed center of mass? I would appreciate if authors can elaborate more on this aspect
3. How strongly does the semantic bias on say actions get transferred from the individual object images? For instance, if the individual object image had a dog simply standing, how difficult/easy would it be to have the dog sitting in the target generation?

---

### Official Review · Reviewer_g5Tm · 2025-11-01

**Soundness:** 3
**Presentation:** 2
**Contribution:** 2
**Rating:** 2
**Confidence:** 4

**Summary:**

* The paper proposes DiDI (Disentangle Denoise Inject), a training-free pipeline that explicitly modifies the initial latent representation to improve Text-to-Image (T2I) alignment, especially under poor initial noise seeds and in multi-subject prompts.
* Extensive experiments demonstrate that DiDI is a plug-and-play method that consistently outperforms existing SOTA approaches and vanilla Stable Diffusion in generating semantically accurate images, particularly those involving multiple subjects.

**Strengths:**

* The proposed DiDI methods directly mitigates common T2I misalignment issues such as subject mixing and subject neglect by addressing their root cause in the initial noise.
* Quantitative analysis shows that DiDI consistently outperforms existing state-of-the-art methods and vanilla Stable Diffusion across various multi-subject datasets.
* Didi is training-free, therefore can support plug-and-play with existing pre-trained T2I diffusion models without requiring expensive fine-tuning.

**Weaknesses:**

* **Reliance on an Outdated and Weak Baseline**: Stable diffusion 1.4 and models of the same family is very weak baseline. Given the rapid advancements in T2I models, which now incorporate stronger text encoders and highly advanced denoiser architectures, the inherent compositional failures (subject mixing/neglect) may be substantially mitigated or even non-existent in more recent, state-of-the-art systems.
  * Self-Cross reported results compared to SD3 medium and SD2. This paper only reports SD 1.4 as the baseline.
  * I test some examples in the paper on a couple recent TTI models like Flux-Kontext-dev and Qwen-Image (open-source), as well as gpt-image-1 and nano-banana (closed-source), and they have no such compositional generation issues at all.
* **Questionable Generalizability to Modern Architectures**: Is DIDI compatible or effective with recently developed model families, such as DiT-based models or Flow Matching models. An analysis of the architectural assumptions DiDI makes (e.g., reliance on U-Net structure or specific cross-attention behavior) and whether it is generalizable to other architectures is needed.
** Limited Scalability and Prompt Complexity Analysis**: The compositional issues highlighted are demonstrated using relatively simple, short multi-subject prompts (e.g., "a bird and a rat"). Current leading T2I models can handle such simple compositions with high fidelity. The true test of a compositional method like DiDI lies in its performance and computational scalability when handling longer, more complex instructions involving multiple subjects (e.g., more than 5 subjects), detailed spatial relationships, and specific attributes. A thorough analysis of DiDI's performance as a function of prompt length and complexity is essential to confirm its real-world utility.

**Questions:**

* Related to Weakness 2: Would DIDI work for more recent, DiT-based diffusion models? What about flow matching models?
* Related to Weakness 3: How scalable is DIDI in terms of prompt length and complexity?

---

### Note · Authors · 2025-11-18

**Comment:**

We respectfully withdraw our submission to address the limitations identified during this review. We appreciate the reviewers' efforts and feedback, which will be used to prepare a more complete manuscript. Thank you.

**Withdrawal Confirmation:**

I have read and agree with the venue's withdrawal policy on behalf of myself and my co-authors.